# Variation in the *HSL* Gene and Its Association with Carcass and Meat Quality Traits in Yak

**DOI:** 10.3390/ani13233720

**Published:** 2023-12-01

**Authors:** Xiangyan Wang, Youpeng Qi, Chune Zhu, Ruifeng Zhou, Zhoume Ruo, Zhidong Zhao, Xiu Liu, Shaobin Li, Fangfang Zhao, Jiqing Wang, Jiang Hu, Bingang Shi

**Affiliations:** 1Gansu Key Laboratory of Herbivorous Animal Biotechnology, College of Animal Science and Technology, Gansu Agricultural University, Lanzhou 730070, China; wxy9242022@163.com (X.W.); 18394174334@163.com (C.Z.); zrf15101229062@163.com (R.Z.); zhaozd@gsau.edu.cn (Z.Z.); liuxiu@gsau.edu.cn (X.L.); lisb@gsau.edu.cn (S.L.); zhaofangfang@gsau.edu.cn (F.Z.); wangjq@gsau.edu.cn (J.W.); 2Maqin County Dawu Town Agricultural and Animal Husbandry Technical Service Station, Guoluo Prefecture 814000, China; 13897154176@163.com

**Keywords:** yaks, *HSL* gene, gene variants, carcass traits, meat quality traits

## Abstract

**Simple Summary:**

Meat quality traits in yak are influenced by molecular genetic regulation, and hormone-sensitive lipase (*HSL*) is a potential gene affecting these traits. *HSL* is responsible for breaking down triacylglycerol in adipose tissue and releasing free fatty acids. Therefore, we used the competitive allele-specific polymerase chain reaction (KASP) technique to detect *HSL* genotypes and analyzed the associations between these genotypes and meat quality traits in yak. The *HSL* gene mutation affected meat tenderness, decreasing the value of Warner Bratzler shear force (WBSF). The highest *HSL* expression was noted in the adipose tissue of yak. These results suggest that the *HSL* gene affects the quality characteristics of yak meat by influencing the fatty acid composition.

**Abstract:**

Hormone-sensitive lipase (*HSL*) is involved in the breakdown of triacylglycerols in adipose tissue, which influences muscle tenderness and juiciness by affecting the intramuscular fat content (IMF). This study analyzed the association between different genotypes and haplotypes of the yak *HSL* gene and carcass and meat quality traits. We used hybridization pool sequencing to detect exon 2, exon 8, and intron 3 variants of the yak *HSL* gene and genotyped 525 Gannan yaks via KASP to analyze the effects of the *HSL* gene variants on the carcass and meat quality traits in yaks. According to the results, the *HSL* gene is highly expressed in yak adipose tissue. Three single nucleotide polymorphisms (SNPs) were identified, with 2 of them located in the coding region and one in the intron region. Variants in the 2 coding regions resulted in amino acid changes. The population had 3 genotypes of GG, AG, and AA, and individuals with the AA genotype had lower WBSF values (*p* < 0.05). The H3H3 haplotype combinations could improve meat tenderness by reducing the WBSF values and the cooking loss rate (CLR) (*p* < 0.05). H1H1 haplotype combinations were associated with the increased drip loss rate (DLR) (*p* < 0.05). The presence of the H1 haplotype was associated the increased CLR in yaks, while that of the H2 haplotype was associated with the decreased DLR in yaks (*p* < 0.05). These results demonstrated that the *HSL* gene may influence the meat quality traits in yaks by affecting the IMF content in muscle tissues. Consequently, the *HSL* gene can possibly be used as a biomarker for improving the meat quality traits in yaks in the future.

## 1. Introduction

Yak (*Bos grunniens*) is a “unique” livestock breed distributed in the Tibetan plateau and its nearby high mountain areas. Yaks have a strong ability to withstand cold and low oxygen and roughage [1]. They can provide biological resources such as meat, milk, and wool for local herders and can provide economic benefits for farmers and herders on the Tibetan Plateau. Yak meat has a high protein content and various crucial vitamins and minerals and is delicious and nutritious [2]. However, the low intramuscular fat (IMF) content, thicker muscle fiber, and coarser texture of yak meat are considered signs of poorer meat quality [3]. The low IMF content is among the major factors limiting consumer acceptance of yak meat. The IMF content is highly valued because it improves meat quality by enhancing flavor, juiciness, and tenderness [4]. However, the IMF content of yak meat is affected by genetics, breeding, environmental nutrition, and other factors [5,6]. Of the aforementioned factors, genetics is of paramount importance because genetic improvement in livestock is permanent and accumulates as it is passed on to future generations [7]. In recent years, marker-assisted selection methods have demonstrated numerous advantages compared with traditional approaches for evaluating specific traits [8]. Therefore, this method must be urgently used to select yaks with higher IMF content and thus promote the genetic improvement of the meat quality traits in yaks.

Hormone-sensitive lipase (*HSL*) is particularly crucial in lipid metabolism and energy homeostasis as it is a rate-limiting enzyme involved in triglyceride hydrolysis [9,10]. *HSL* gene activation is regulated by the cAMP cascade system, starting with the combination of the hormone with its specific cytosolic receptor, forming a hormone receptor compound [11]. This compound then interacts with the Gs protein, freeing the alpha subunit of the activated adenylate cycle. The activated cycle catalyzes cAMP production from ATP. The modulatory substrates of protein kinase A bind to cAMP, releasing and activating the catalytic subunit [12,13]. Hormone-sensitive lipase is among the intracellular proteins phosphorylated by activated kinases, and the phosphorylated adipocyte *HSL* gene is the active form of the *HSL* gene. The activated *HSL* gene hydrolyzes the first fatty acids in triacylglycerols stored in adipose tissue, leading to the release of free fatty acids, which affects the fat deposition and meat quality. The *HSL* gene, believed to be a ‘‘mosaic gene,” is situated on chromosome 18 in cattle and comprises 9 exons. Each of these exons encodes a distinct structural domain and can be altered by other exons to create a chimeric product. *HSL* is a multifunctional gene that is highly expressed in adipose tissue [14]. *HSL* is indispensable for lipid metabolism [15] and triglyceride hydrolysis in fat cells is catalyzed by 2 rate-limiting enzymes via 3 consecutive reactions in which the *HSL* gene can hydrolyze triacylglycerol, diacylglycerol, and monoacylglycerol in adipose tissue into free fatty acids. Therefore, the *HSL* gene is known as the gatekeeper in lipid metabolism. *HSL* gene variants are associated with the body mass index and risk of obesity in humans [16,17]. Furthermore, the *HSL* gene plays a role in insulin resistance, diglyceride levels, and fatty acid composition [18]. The identification of 2 SNPs within the *HSL* gene, specifically in intron 2 and the promoter region, has revealed associations with blood glucose-related phenotypes in an obese population [19]. *HSL* gene overexpression in bovine fetal fibroblasts negatively regulates adipogenic transcription factors [20]. Since the *HSL* gene is involved in free fatty acid metabolism in the body, *HSL* gene variants affects the fatty acid composition of meat, and thus, its quality [21]. The *HSL* gene influences carcass and meat characteristics, including pH and fatty acid composition of cattle [22,23]. G/A variants were found in exon 1 of the porcine *HSL* gene and significant differences in REA were noted between the AG and GG genotypes [24]. *HSL* gene variants are related to the IMF content of Qinchuan cattle and Nanyang cattle [25].

Currently, no study has examined *HSL* gene variation in yaks. Therefore, it is necessary to detect variation in the *HSL* gene and its effect on the meat quality traits in yaks. In this study, we examined the polymorphism of the *HSL* gene in yak, assessed the effects of variations in this gene on the carcass and meat quality traits, and enriched the molecular genetic data of the economic traits in yaks.

## 2. Materials and Methods

The animal study was approved by the Animal Care Committee at Gansu Agricultural University (approval number GAU-LC-2020-056). All animals in this study were approved by the Animal Ethics Committee of Gansu Agricultural University (approval number 2006-398).

### 2.1. Animals and Sample Collection

We selected 525 Gannan yaks, aged 3 to 7 years (351 males, 174 females), residing in the Gannan Tibetan Autonomous Prefecture, Gansu Province, China. Gender and age information for each yak was recorded. Prior to slaughter, the yaks were fasted for 24 h, with access to ample water. Blood samples were collected from the jugular of each yak in 10 mL sodium heparin tubes (Thermo Fisher Scientific, Waltham, MA, USA) and stored at −80 °C for genomic DNA extraction.

Post-slaughter, the longest dorsal muscle was collected of each yak for meat quality determination. Additionally, 3 yaks were chosen and tissues (heart, liver, spleen, kidney, longest dorsal muscle, large intestine, rumen, testis, small intestine, subcutaneous fat, and lungs) were rapidly collected and stored at −80 °C for further analysis.

### 2.2. RNA Extraction and qRT-PCR Analysis

Total RNA was extracted using the Trizol reagent according to the manufacturer’s instructions. The extracted RNA was reverse transcribed to cDNA using the PrimeScriptTM RT kit and gDNA reagent. The reaction system for qRT-PCR comprised 0.8 µL of cDNA, 0.25 µM of each primer, 10.0 µL of AceQ qPCR SYBR^®^ Green Premix (Vazyme, Nanjing, China), and 0.4 µL of ROX Reference Dye 2, and the volume was fixed to 20 µL with free water. The reaction steps were pre-denaturation at 95 °C for 30 s; denaturation 5 s; extension at 95 °C, 30 s at 60 °C; melting 5 s at 95 °C, 1 min at 60 °C; and cooling at 50 °C for 30 s.

### 2.3. Meat Quality Trait Measurements

The carcass and meat quality traits were determined according to GB/T17238-1998 “China Fresh and Cold Beef Division Standards. Five traits were measured and calculated: WBSF (kg), CLR (%), DLR (%), REA (cm^2^), and HCW (kg). The HCW was measured for each Gannan yak immediately after the animal was slaughtered. At 48 h after slaughter, between the 12th and 13th thoracic ribs, the interface of the longissimus dorsi muscle representing the REA was traced with sulfuric acid paper and estimated using a product meter. Additionally, samples of the longissimus dorsi muscles were collected from the 11th and 12th thoracic ribs of the right carcass and stored at −18 °C for measuring WBSF, DLR, and CLR. DLR was measured according to Liu et al.’s method [26], WBSF was measured according to Shackelford et al.’s method [27], and CLR was measured according to Honikel’s method [28].

### 2.4. Genomics DNA Isolation

Genomics DNA was extracted from the blood by using a blood DNA kit (TIANamp Genomic DNA Purification Kit; Tianen Biotech, Beijing, China) according to the manufacturer’s instructions. Genomics DNA was stored at −80 °C.

### 2.5. PCR Amplification and SNP Identification

Primers for 3 regions in intron 3 and exons 2 and 8 of the *HSL* gene were designed according to the yak *HSL* gene sequence (GenBank No. XM_014482659.1) by using primer5 software (version 5.0). Based on the mRNA sequences of yak *HSL* and *β-actin*, 2 qPCR primer pairs were designed to amplify the mRNA sequences of yak *HSL* and *β-actin* (GenBank accession numbers No. XM_014482659.1 and DQ838049.1, respectively) (Table 1). The primers were synthesized by Beijing Qingke Biotechnology Co. (Hangzhou, China). The genomic DNA of 30 yaks was used to amplify the 3 *HSL* gene regions. The mixing system consisted of a 20 μL mixture, including 10 μL of 2× Taq PCR master mix (Tiangen Biotech, Beijing, China), a 2 μL DNA template, 0.8 μL upstream and downstream primers, and 6.4 μL ddH_2_O. The thermal cycling reaction procedure was as follows: denaturation at 95 °C for 5 min, 30 cycles at 94 °C, annealing at 55 °C for 30 s, primer extension at 72 °C for 30 s, and a final extension at 72 °C for 10 min. The PCR products were sent to ShengGong Biotech (Shanghai, China) for sequencing. The sequencing results were aligned by BLAST to detect SNPs.

### 2.6. Genotyping

The KASP genotyping assay was conducted by Gentiles Biotechnology Co., Ltd. (Wuhan, China). Table 2 presents the primer information for genotyping. The method applied was as follows: two forward and one reverse primers with 2 universal tags and SNP sites constituting the primer mix, two detection primers with different fluorescence signals constitute the Master mix, and the SNP sites were detected fluorescently via 3 PCR reactions. PCR reaction I: The DNA template was denatured; bound to the matching KASP primers; and following annealing and extension, the detection primer sequence was added. PCR reaction II: A complementary strand of the allele-specific terminal sequence was synthesized. PCR reaction III: The detection primer corresponding to the specific sequence grew exponentially during PCR and the fluorescence signal was generated and detected. After the reaction, the fluorescence data were read using a microplate reader with a fluorescence resonance energy transfer and genotyping maps were generated using LGC-OMEGA software (version 2.22).

### 2.7. Statistical Analyses

The genotype frequency, allele frequency, homozygosity (Ho), heterozygosity (He), were calculated and Hardy–Weinberg equilibrium (HWE) was tested by referring to Nei et al.’s method [29]. The number of effective allele (Ne) and polymorphic information content (PIC) of yak *HSL* gene SNPs were calculated using the formula put forward by Botstein et al. [30]. The haplotypes and linkage disequilibrium (LD) of SNPs were analyzed using Haploview 4.2 software.

The general linear model (GLM) of SPSS26.0 was used to determine the association of the genotypes, haplotype combinations, and haplotypes with the carcass and meat quality traits. The GLM was as follows: Y_ijkl_ = μ + G_i_ + A_j_ + S_k_ + F_l_ + e_ijkl_, Y_ijkl_ = μ + D_i_ + A_j_ + S_k_ + F_l_ + e_ijkl_, Y_ijkl_ = μ + H_i_ + A_j_ + S_k_ + F_l_ + e_ijkl_, where Y represented the trait phenotypic value, μ represented the overall mean, G_i_ represented the genotype effect, D_i_ represented the haplotype combination effect, H_i_ represented the haplotype effect, A_j_ represented the age effect, S_k_ represented the sex effect, F_l_ represented the field effect, and e_ijkl_ is the random error.

Any haplotype with an associated *p* < 0.2 in the single haplotype model and therefore could have potentially influenced the trait was included in the model. This allowed the identification of independent haplotype effects.

## 3. Results

### 3.1. Tissue Expression of HSL

The relative *HSL* gene expression was identified in 11 tissues of yaks. The *HSL* gene was commonly expressed in various tissues (Figure 1). The highest *HSL* expression was in the adipose tissue, the heart, rumen, kidney, testis, and liver, while the *HSL* gene was expressed to a lesser extent in the spleen, lung, small intestine, and large intestine.

### 3.2. Variation in the HSL Gene in Yaks

3 amplified fragments of the *HSL* gene were found to be polymorphic via DNA detection (Figure 2). *HSL* gene sequencing in the 30 yaks revealed that the SNPs were g.78944 (SNP1) in exon 2, and g.81959 (SNP2) in intron 3, g.84209 (SNP3) in exon 8, respectively. The KASP genotyping 3 genotypes for each of the 3 SNPs: AA, AG, and GG (Figure 3). The frequency of the GG genotype was higher than that of the AA genotype. Moreover, the SNP1 base shift from G to A caused the amino acid to change from glycine to arginine, and the SNP3 grade transition from G to A led to the amino acid change from arginine to glutamine. The population genetic analysis revealed that these SNPs were in HWE in the yak population (Table 3).

### 3.3. LD Analyses of the HSL Gene

For SNP1, SNP2, and SNP3 in *HSL*, LD was analyzed using Haploview. D′ represents the LD level of the population and r^2^ is used to predict the determination coefficient of SNPs. Table 3 presents the results of the LD analysis. The 3 SNPs were in a strong linkage state (Figure 4). The 3 haplotypes were constructed in the yak populations and formed six haplotype combinations with frequencies of >0.03 (Table 4).

Haplotype structural analysis revealed that the yaks had 3 major haplotypes, with the highest frequency of H1 (-GGG, 0.428), followed by H2 (-GAA-, 0.376) and H3 (-AGG-, 0.194) (Table 5).

### 3.4. Effects of the HSL Genotype on Meat Quality Traits and Carcass Traits

The association analysis between the genotype and meat and carcass traits revealed that individuals with GG and AG genotypes in SNP1 had higher WBSF and CLR values than those with AA genotypes (*p* < 0.05) (Table 6). The individuals with AG and GG genotypes at SNP2 and SNP3 had higher WBSF values, whereas individuals with the GG genotype had higher DLR means than those with the AA and AG genotypes (*p* < 0.05). No significant effects of different genotypes on REA or HCW were observed.

### 3.5. Association Analysis of HSL Haplotype Combinations with Meat Quality Traits and Carcass Traits

Significant associations between *HSL* haplotypes and key meat and carcass quality traits were identified in our study. Specifically, the presence of the H1 haplotype was found to be associated with a notable increase in CLR value, reaching statistical significance (*p* < 0.05) as detailed in Table 7. Importantly, these associations remained significant even after accounting for the presence of other haplotypes, such as H2, within the model. Furthermore, when considering the simultaneous presence of haplotypes H1 and H2 in the model, the H3 haplotype exhibited a significant association with an elevated CLR value (*p* < 0.05). Conversely, the presence of H2 haplotypes was linked to a reduction in DLR, demonstrating statistical significance (*p* < 0.05).

An association analysis conducted between the haplotype combinations and carcass and meat quality traits unveiled that the WBSF values for H1H1, H1H2, H1H3, and H2H3 were higher than that for H3H3 (Table 8). The CLR value of H3H3 was lower than that of other haplotype combinations (*p* < 0.05), and the DLR value of H1H1 was higher than those of H1H2 and H3H3 (*p* < 0.05).

## 4. Discussion

The slow growth of yak, inefficient fat deposition, poor tenderness, and rough texture are not favorable for producing and developing yak products in the Tibetan Plateau and thus affect their economic value. Therefore, enhancing the growth performance and meat quality traits of yaks is currently the main direction of yak breeding. IMF can improve meat tenderness by reducing the intramuscular connective tissue structure [31]. Therefore, improving yak meat tenderness in terms of its IMF content is an effective method for enhancing yak meat quality. The IMF content was determined on the basis of fat acid composition and lipid metabolism [32], where many lipid metabolism genes probably have a role in IMF content production [33]. The *HSL* gene was one of the rate-limiting enzymes involved in regulating fatty acid metabolism [9].

The *HSL* gene is highly expressed in adipose and muscle tissues of humans and mice [21,34]. We found that yak *HSL* gene expression was the highest in the adipose tissue, followed by the heart, rumen, liver, dorsal muscle, testis, and liver, which is consistent with the results of Yeaman et al. [35]. Adipose tissue is an important tissue for storing energy, when the body is in a state of energy deficit, hunger, excitement, etc., adrenaline secretion is stimulated. This hormone activates adenylate cyclase in the cell membrane, thereby accelerating the synthesis of cyclic adenosine monophosphate (cAMP). cAMP activates protein kinase A, which in turn activates the *HSL* gene. Each of these signal transduction processes has a signal amplification effect, which results in the immense breakdown of triglycerides stored in adipose tissue into triglycerides and free fatty acids. Subsequently, triglycerides are hydrolyzed by diglyceride lipase and monoglyceride lipase to form glycerol and fatty acids, which are then transported to other tissues to provide energy via oxidation. This potentially clarifies the reason for the elevated *HSL* gene expression observed in adipose tissue. Moreover, the highest *HSL* gene expression in adipose tissue reinforces that the *HSL* gene has a role in fatty acid metabolism, and so, we hypothesized that the *HSL* gene is associated with IMF deposition in yaks.

The candidate gene approach is among many methods of relating phenotypes and genotypes and is the most advantageous strategy for genetically enhancing the quantitative traits in ruminants [36,37]. Nowadays, an increasing number of observations have demonstrated that the *HSL* gene is an outstanding candidate for the meat quality traits [22]. This gene encodes for an intracellular neutral lipase that hydrolyzes various estrogens and is involved in mobilizing free fatty acids in living bodies; therefore, changes in the sequence of this gene may alter the fatty acid composition of tissues [38]. The present study is the first to reveal the relationship between the variation in the *HSL* gene and meat and carcass traits in yaks. We have identified here 3 SNPs of the *HSL* gene, with two of them located in the exon 2 and intron3 regions. The g.78944G>A and g.81959G>A variants in exon 2 led to amino acid changes from glycine to arginine and from serine to asparagine, respectively. Therefore, variations in the amino acid coding region may cause alterations in the HSL protein structures that result in functional *HSL* lipid metabolism, thereby influencing the meat composition and altering the meat quality traits [38,39]. In the current study, individuals with the AA genotype had lower WBSF values than those with the GG phenotype, possibly because these amino acid mutation variants alter the function of the yak HSL proteins. This then affects the contribution of *HSL* to fat metabolism, IMF content in the muscles, and ultimately meat tenderness in yaks. Thus, selecting and breeding individuals with the AA genotype is important to improve the meat quality of yaks in the future. Several polymorphisms have recently been identified in the *HSL* gene of other species, and these polymorphisms are significantly associated with other economic traits. For example, two mutations were found in exons 1 and 8 of the *HSL* gene in Simmental cattle, as well as mutant variants in the *HSL* gene were found to be related to their meat characteristics, such as pH and fatty acid composition [22]. A new SNP (s974514528) identified in the 5′ non-translated region of the *HSL* gene was linked to altered oleic acid and total monounsaturated fatty acid contents in the meat of crossbred cattle [38]. Five novel variants in the 5′ terminal sequence of the *HSL* gene can alter the binding of transcription factors and therefore of transcription levels, ultimately modifying the fatty acid composition of Chinese Simmental cattle [20]. Moreover, *HSL* gene expression was positively correlated with the IMF content of the longest bovine dorsal muscle [33]. Yang et al. reported that 3 single-nucleotide variants in the fifth intron of the *HSL* gene in Suffolk sheep with a nonsynonymous mutation in exon 9 were identified, and the fifth intron variation was associated with REA depth and width in the sheep, whereas the variant in exon 9 was not associated with the growth or carcass traits [40]. The aforementioned results suggest that genetic variation in the *HSL* gene may influence the fat composition and IMF content, thereby affecting the meat quality traits, which is similar to our findings. Consequently, we speculated that variation in yak *HSL* affects meat tenderness by regulating the IMF content. Furthermore, SNPs in introns influence gene expression and the mRNA translation efficiency in many eukaryotes by modulating transcription rates, nuclear export, and transcript stability [41]. One or more introns are also required for the optimal expression of many genes [42]. We here found that variants in intron 3 of the yak *HSL* gene possibly affect gene expression or transcription and influence fat deposition, which in turn affects yak meat tenderness.

Genetic variation is crucial because of its abundance and association with quantitative trait loci and in remodeling proteins generated from these loci [43]. Meat quality traits are quantitative traits under the control of multiple genes and their phenotypic changes are affected by many single mutations sites. Moreover, compared with single mutations, the cascading influence of multiple mutation sites has a greater impact on the changes in biological phenotypic traits [44]. Since haplotype combination analysis considers non-allelic interactions and LD between multiple variant loci, this combination analysis has a better statistical power than statistical analysis between alleles [45]. Haplotype analysis has recently been a field of study for complex genetic phenotypes [46]. A haplotype can be described as a group of SNPs tightly linked on a single chromosome and inherited as a unit. The haplotype knowledge of multiple SNPs in a gene would offer more crucial information on genotype–phenotype associations than a single potential SNP. For example, in pigs, the IMF content was higher in individuals with Hap 4/5 and Hap 5/5 combinations of the *SIRT1* gene than in those with other haplotype combinations [47]. In a Tibetan sheep population, Zhao et al. found that individuals with the *HIF* gene having H1H3 had a higher oxygen transport capacity than those with other haplotype combinations [48]. Our statistical results revealed that individuals with H3H3 were significantly associated with WBSF than those with other haplotypes. H3H3 haplotypes can be screened as molecular markers for combined genotypes towards meat quality improvement in yaks. Additionally, SNP2 and SNP3 loci are in LD and more pronounced phenotypic changes in organisms, further confirming that H3H3 had a significant effect on WBSF values compared with other haplotype combinations. 

In conclusion, the aforementioned findings suggest that the *HSL* gene is one of the crucial genes affecting meat quality traits in yaks. Therefore, the use of marker-assisted selection in larger populations will be necessary to improve meat quality in yaks. We found that 3 SNPs in the *HSL* gene had no effect on yak carcass traits, and so, further screening of candidate genes for yak carcass traits is required to enhance basic research for improving the yak production performance.

## 5. Conclusions

3 variants were found in the *HSL* gene of yak. The association analysis involving the polymorphisms (g.78944G>A, g.84209G>A, and g.81959G>A) revealed significant effects on the WBSF and CLR values. The haplotypes H2H2 and H2H3 significantly affected the WBSF values of yak. The *HSL* gene may have regulatory roles that influence meat quality traits in yaks. The *HSL* gene may be mainly involved in regulating the IMF content. The present study may offer new opportunities for further research on potential genetic factors affecting meat quality traits in yaks.

## Figures and Tables

**Figure 1 animals-13-03720-f001:**
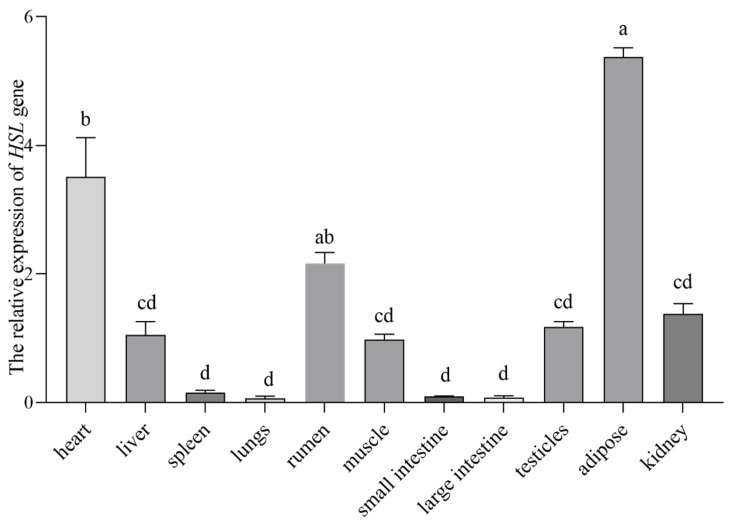
Tissue expression analysis of yak *HSL* mRNA. Different lowercase letters indicate significant differences in expression in different tissues.

**Figure 2 animals-13-03720-f002:**
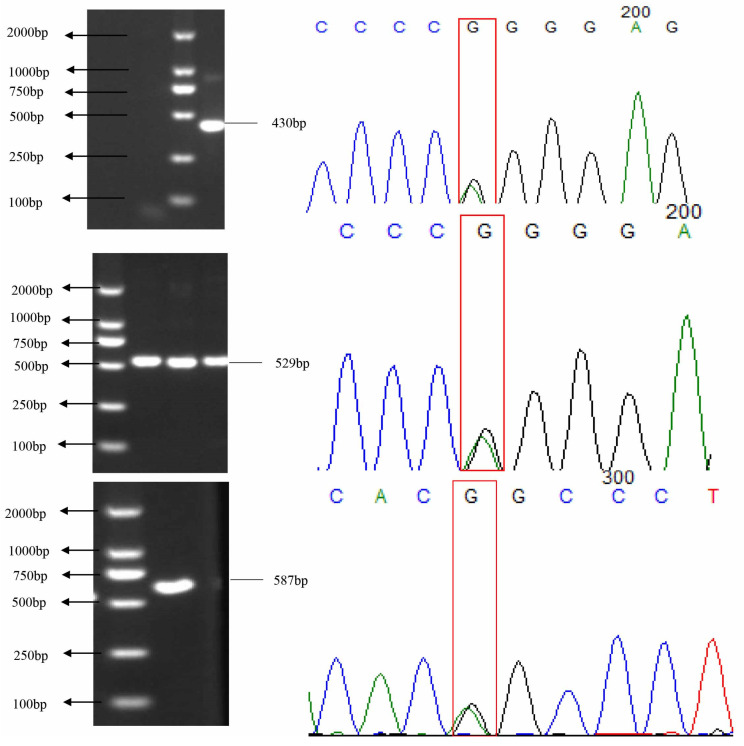
Amplification and sequencing results of exons 2 and 8 and intron 3 of the yak *HSL* gene. Red boxes indicate SNP sites.

**Figure 3 animals-13-03720-f003:**
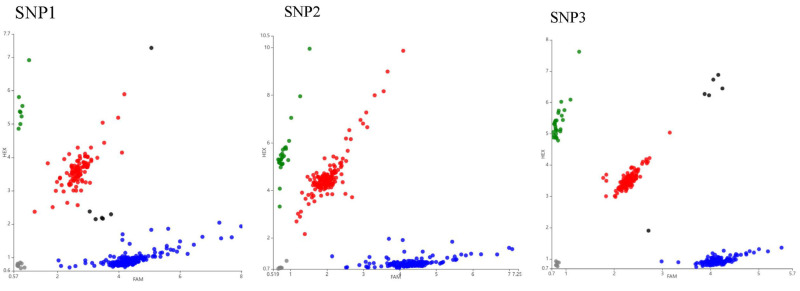
Results of the KASP typing assay for 3 loci of the *HSL* gene of yaks. Red, blue, and green dots indicate AG, GG, and AA genotypes, respectively.

**Figure 4 animals-13-03720-f004:**
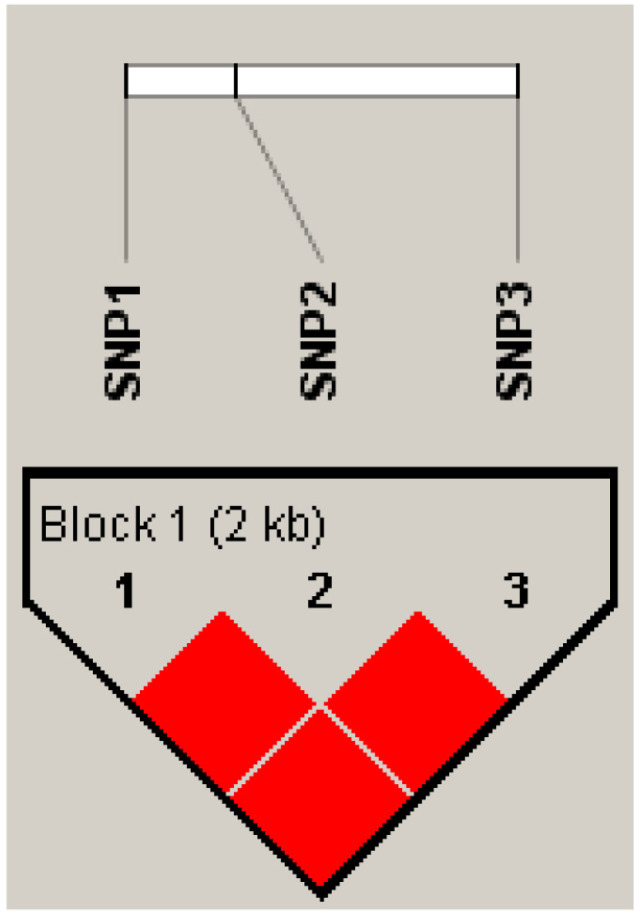
LD plots for the 3 SNPs of the *HSL* gene. Red color represents high paired D’ values.

**Table 1 animals-13-03720-t001:** Primer information of the *HSL* gene.

Gene	Region	Primer Sequence (5′–3′)	Amplicon Size (bp)	Primer Usage
*HSL*	Exon 2	F: TGTGAGTCAGGGGGAGGAGAGGGTA R: AGATGCTGCGGCGGTTGGAG	430	PCR
	Exon 8	F: CTCTCAGCGTGCTCTCCAAGTGCGT R: TGTGGACTAAGGCTTTATGG	525	PCR
	Intron 3	F: TGGACTGGTGTCCTTCGGGGAGCAC R: CGAGGTCAGAGGCATTTCA	587	PCR
	CDS	F: CGGGACCGCGGAAAATTGAT R: GCGCCTTTGACTTTTGGACC	169	qRT-PCR
*β-actin*	CDS	F: AGCCTTCCTTCCTGGGCATGGA R: GGACAGCACCGTGTTGGCGTAGA	113	qRT-PCR

**Table 2 animals-13-03720-t002:** Primer information for SNP genotype of *HSL* gene.

Region	Primer Sequence (5′–3′)	Primer_AlleleGG (5′–3′)	Primer_AlleleAA (5′–3′)
Exon 2	TGACCCTGGCAG AGGACAAC	GAAGGTGACCAAGTTCATGCT CGGGCCGTCTCCCC	GAAGGTCGGAGTCAACGGA TTCCGGGCCGTCTCCCT
Exon 8	CGACTCAGACAGAAGGCG	GAAGGTGACCAAGTTCATGCTCTCGCAAGAGCAGGGCC	GAAGGTCGGAGTCAACGGATTG TCTCGCAAGAGCAGGGCT
Intron 3	CCCCTACCCTTCCTGTCCCT	GAAGGTGACCAAGTTCATGCTCCCCTGGTGGCCTTGC	GAAGGTCGGAGTCAACGGATTGCCCCTGGTGGCCTTGT

**Table 3 animals-13-03720-t003:** Population genetic analysis of the *HSL* gene.

SNPs	Genotype Frequency			Allele Frequency		Genetic Polymorphism				HWE
	Genotype	Number	Frequency (%)	Allele	Frequency (%)	Ne	Ho	He	PIC	
Exon 2	GG	353	67.37	G	82.00	1.41	0.71	0.29	0.25	*p* > 0.05
	AG	155	29.58	A	18.00					
	AA	16	3.05							
Intron 3	GG	256	48.85	G	71.00	1.71	0.58	0.42	0.33	*p* > 0.05
	AG	227	43.32	A	29.00					
	AA	46	7.82							
Exon 8	GG	257	48.12	G	70.00	1.71	0.58	0.42	0.33	*p* > 0.05
	AG	230	44.75	A	30.00					
	AA	38	7.11							

**Table 4 animals-13-03720-t004:** LD analysis of the *HSL* gene.

	SNP1	SNP2	SNP3
SNP2	0.08		
SNP3	0.08	1	

**Table 5 animals-13-03720-t005:** Haplotypes and haplotype combinations of triple SNPs in the *HSL* gene.

Haplotypes	SNP1	SNP2	SNP3	Frequency/%	Haplotype Combinations	Frequency/%
H1	G	G	G	42.8	H1H1	28.2
H2	G	A	A	37.6	H1H2	30.1
H3	A	G	G	19.4	H2H3	15.0
					H2H2	7.2
					H1H3	15.4
					H3H3	3.9

**Table 6 animals-13-03720-t006:** Association between genotypes of the *HSL* gene and carcass and meat quality traits of yaks.

SNPs	Genotype	Meat Quality					Carcass Quality	
		*n*	WBSF	CLR	DLR	REA	*n*	HCW
SNP1	GG	353	5.58 ± 0.11 ^a^	65.97 ± 0.42 ^a^	21.22 ± 0.40	32.49 ± 0.61	123	105.65 ± 3.97
	AG	155	5.68 ± 0.14 ^a^	66.21 ± 0.56 ^a^	21.64 ± 0.554	31.95 ± 0.82	46	105.93 ± 5.27
	AA	16	4.06 ± 0.45 ^b^	58.13 ± 1.74 ^b^	20.44 ± 0.68	30.92 ± 2.55	4	101.20 ± 17.88
	*p*		0.002	<0.001	0.608	0.687		0.967
SNP2	GG	256	5.50 ± 0.12 ^a^	65.36 ± 0.47	22.44 ± 0.44 ^a^	32.50 ± 0.68	80	105.30 ± 4.27
	AG	227	5.74 ± 0.12 ^a^	65.45 ± 0.48	21.04 ± 0.46 ^b^	32.01 ± 0.70	77	106.56 ± 4.42
	AA	41	4.97 ± 0.31 ^b^	65.42 ± 1.22	21.61 ± 1.16 ^b^	33.47 ± 1.76	16	102.63 ± 10.34
	*p*		0.022	0.187	0.024	0.627		0.924
SNP3	GG	257	5.51 ± 0.12 ^a^	66.41 ± 0.47	22.43 ± 0.44 ^a^	32.54 ± 0.67	80	105.30 ± 4.27
	AG	230	5.76 ± 0.12 ^a^	65.43 ± 0.48	21.19 ± 0.46 ^b^	32.01 ± 0.70	77	106.56 ± 4.42
	AA	38	4.99 ± 0.31 ^b^	65.46 ± 1.22	21.81 ± 1.16 ^b^	33.51 ± 1.76	16	102.63 ± 10.34
	*p*		0.020	0.143	0.038	0.605		0.924

Note: Values in the table are mean ± standard errors; the different letters on the shoulder of data means in the same column with different letters are significantly different (*p* < 0.05). HCW indicates hot carcass weight, REA indicates rid eye area.

**Table 7 animals-13-03720-t007:** Association of the presence/absence of *HSL* haplotypes with carcass and meat quality traits in yak.

			*n*		Single-Haplotype Model		*p*
Traits	Haplotype	Other Haplotypes in Model	Present	Absent	Present	Absent	
WBSF	H1		396	89	5.64 ± 0.10	5.45 ± 0.17	0.251
	H2		238	247	5.69 ± 0.12	5.51 ± 0.12	0.151
	H3		152	333	5.60 ± 0.11	5.60 ± 0.14	0.985
	H1	H2	328	247	5.66 ± 0.11	5.34 ± 0.17	0.071
	H3	H2	152	333	5.62 ± 0.14	5.59 ± 0.111	0.843
CLR	H1		396	89	65.71 ± 0.40 ^a^	63.83 ± 0.63 ^b^	0.003
	H2		238	247	65.00 ± 0.46	65.69 ± 0.45	0.151
	H3		152	333	65.54 ± 0.54	65.28 ± 0.41	0.617
	H1	H2	396	89	65.69 ± 0.40 ^a^	63.89 ± 0.66 ^b^	0.008
	H2	H1	238	247	64.71 ± 0.47	64.88 ± 0.54	0.742
	H3	H1H2	152	333	65.48 ± 0.54 ^a^	64.13 ± 0.53 ^b^	0.033
DLR	H1		396	89	21.88 ± 0.40	22.773 ± 0.58	0.789
	H2		238	247	21.25 ± 0.44 ^a^	22.43 ± 0.44 ^b^	0.012
	H3		152	333	21.67 ± 0.53	21.91 ± 0.53	0.650
REA	H1		396	89	32.51 ± 0.60	31.66 ± 0.94	0.357
	H2		238	247	32.15 ± 0.68	32.55 ± 0.67	0.579
	H3		152	333	32.92 ± 0.81	32.92 ± 0.81	0.452
HCW	H1		124	30	106.34 ± 3.67	103.20 ± 5.97	0.617
	H2		79	75	106.06 ± 4.19	105.24 ± 4.25	0.870
	H3		44	110	105.64 ± 5.13	105.67 ± 3.78	0.996

Note: the different letters on the shoulder of data means in the same line with different letters are significantly different (*p* < 0.05). A single haplotype indicates that one haplotype is the most independent variable, other haplotypes indicated that single haplotypes associated with traits with *p* < 0.2 in a single haplotype are fitted to fixed factors.

**Table 8 animals-13-03720-t008:** Association of *HSL* haplotypes with meat quality traits and carcass traits in yaks.

Diplotypes	Meat Quality					Carcass Quality	
	*n*	WBSF	CLR	DLR	REA	*n*	HCW
H1H1	144	5.54 ± 0.13 ^a^	65.68 ± 0.50 ^a^	22.88 ± 0.49 ^a^	32.56 ± 0.76	48	104.50 ± 5.01
H1H2	145	5.71 ± 0.13 ^a^	65.18 ± 0.51 ^a^	20.88 ± 0.51 ^b^	32.32 ± 0.78	55	107.51 ± 5.02
H1H3	75	5.58 ± 0.18 ^a^	66.59 ± 0.66 ^a^	21.61 ± 0.66 ^ab^	32.61 ± 1.01	29	107.68 ± 6.87
H2H2	35	4.97 ± 0.31 ^ab^	62.80 ± 1.15 ^a^	21.74 ± 1.15 ^ab^	33.48 ± 1.77	15	102.79 ± 10.44
H2H3	69	5.84 ± 0.20 ^a^	65.22 ± 0.74 ^a^	21.78 ± 0.74 ^ab^	31.06 ± 1.37	20	103.80 ± 7.68
H3H3	21	4.06 ± 0.45 ^b^	57.84 ± 1.67 ^a^	20.62 ± 1.66 ^b^	30.95 ± 2.55	6	101.25 ± 18.04
*p*		0.004	<0.001	0.011	0.856		0.989

Note: the different letters on the shoulder of data means in the same column with different letters are significantly different (*p* < 0.05).

## Data Availability

Data are contained within the article

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
