# Peer review of "Variation in the HSL Gene and Its Association with Carcass and Meat Quality Traits in Yak"

_animals, 2023, doi:10.3390/ani13233720_

Round 1

Reviewer 1 Report

Comments and Suggestions for Authors

1 Suggest replacing the photo in Figure 2, which has poor clarity, and adding strip lab.

2 At the beginning of the new sentence, it is recommended that the HSL be replaced with the full name of the gene.

3 Attention to the problem of writing on line 132.

4 In the discussion section, it has been suggested that haplotype combinations are more influential than haplotypes and that the correlation between HSL haplotypes and meat quality traits should be discussed in depth.

5. To determine whether the location of the HSL gene on the yak chromosome is consistent with that of the common cow, since the subjects of this study were yaks.

6. Suggested revision of line 89, were, to read was.

7 Change longest dorsal muscle in line 134 to longissimus dorsi muscles.

8 Reorganize the language in line 82.

9 mutations change to variants, full text unification.

10 Check the original tenses throughout the text.

11 Check for italicized genes throughout the text.

12 Proposed delete of 17 lines.

13 Simplify the discussion of the first paragraph.

14 Start of new sentence, suggest "It has been reported that the ....

15 Based on the results of the linkage disequilibrium analysis showing that SNP2 and SNP3 are in full linkage, it is suggested that this part of the discussion be included in the discussion section

Comments on the Quality of English Language

Some sentences need polishing

Author Response

Dear reviewer:

Thank you for your review and suggestions on our manuscript. Those comments are all valuable and very helpful for revising and improving our paper. We have studied comments carefully and have made correction which we hope meet with approval. The comments are laid out below in italicized font and specific concerns have been numbered. Our response is given in red font and changes/additions to the manuscript are given in the red text. The detailed corrections are listed below.

Q1: Suggest replacing the photo in Figure 2, which has poor clarity, and adding strip lab.

Thank you for carefully reading our manuscript and giving us valuable suggestion.

Q2: At the beginning of the new sentence, it is recommended that the HSL be replaced with the full name of the gene

Thank you for carefully reading our manuscript and giving us valuable suggestion.

Corrected.

Q3: Attention to the problem of writing on line 132.

Thank you for carefully reading our manuscript and giving us valuable suggestion.

Corrected.

Q4:  In the discussion section, it has been suggested that haplotype combinations are more influential than haplotypes and that the correlation between HSL haplotypes and meat quality traits should be discussed in depth.

Thank you for carefully reading our manuscript and giving us valuable suggestion.

Corrected.

Q5: To determine whether the location of the HSL gene on the yak chromosome is consistent with that of the common cow, since the subjects of this study were yaks.

Thank you for carefully reading our manuscript and giving us valuable suggestion.

According to the NCBI data, the HSL gene has not been identified on which chromosome, so we referenced the chromosomal location of the HSL gene in common cattle

Q6:  Suggested revision of line 89, were, to read was.

Thank you for carefully reading our manuscript and giving us valuable suggestion.

Corrected.

Q7:  Change longest dorsal muscle in line 134 to longissimus dorsi muscles.

Thank you for carefully reading our manuscript and giving us valuable suggestion.

Corrected.

Q8:  Reorganize the language in line 82.

Thank you for carefully reading our manuscript and giving us valuable suggestion.

corrected

Q9:  mutations change to variants, full text unification.

Thank you for carefully reading our manuscript and giving us valuable suggestion.

Corrected.

Q10:  Check the original tenses throughout the text.

Thank you for carefully reading our manuscript and giving us valuable suggestion.

corrected

Q 11: Check for italicized genes throughout the text.

Thank you for carefully reading our manuscript and giving us valuable suggestion.

Corrected.

Q 12: Proposed delete of 17 lines.

Thank you for carefully reading our manuscript and giving us valuable suggestion.

Corrected.

Q 13: Simplify the discussion of the first paragraph.

Thank you for carefully reading our manuscript and giving us valuable suggestion.

Corrected

Q 14: Start of new sentence, suggest "It has been reported that the ....

Thank you for carefully reading our manuscript and giving us valuable suggestion.

Corrected

Q 15: Based on the results of the linkage disequilibrium analysis showing that SNP2 and SNP3 are in full linkage, it is suggested that this part of the discussion be included in the discussion section.

Thank you for carefully reading our manuscript and giving us valuable suggestion.

Corrected.

Reviewer 2 Report

Comments and Suggestions for Authors

The manuscript entitled " Sequence and haplotypes of HSL gene and their association with 2 carcass and meat quality traits in yak" The aim of the study is to identify the polymorphism of the HSL gene, and then they have evaluated the effects of changes in this gene on the quality characteristics of the carcass and meat, and finally, the molecular genetic data of economic traits in cattle have been analyzed gene ontology. It is an interesting and up-to-date study. The overall intention of this submission is a good one. 

 I have provided my comments as follows.

 Q1-The author should mention in the paper that she designed the primers used or obtained them from another paper.

Q2-  Authors should add limitations and strengths in the introduction section.

Author Response

Response to reviewer 2:

Dear reviewer:

Thank you for your review and suggestions on our manuscript. Those comments are all valuable and very helpful for revising and improving our paper. We have studied comments carefully and have made correction which we hope meet with approval. The comments are laid out below in italicized font and specific concerns have been numbered. Our response is given in red font and changes/additions to the manuscript are given in the red text. The detailed corrections are listed below.

 Q1-The author should mention in the paper that she designed the primers used or obtained them from another paper.

Thank you for carefully reading our manuscript and giving us valuable suggestion.

We have supplemented the article with software for primer design.

 Q2 Authors should add limitations and strengths in the introduction section.

Thank you for carefully reading our manuscript and giving us valuable suggestion.

We have already added in the introduction the advantages of molecular genetic markers

Reviewer 3 Report

Comments and Suggestions for Authors

The manuscript deals with to  sequence and haplotypes of HSL gene and their association with  carcass and meat quality traits in yak. Topics concerning indigenous species are always welcome and should be more exploited in order to aid their preservation. I feel that the topic of this research paper is relevant for the journal. However, I also have a few reservations about the manuscript.  I've given my comments and suggestions below, which I believe will improve this paper.

 Comments:

Material and Methods

-          Part 2.5 „PCR amplification and SNP identification“:  

1. What software was used to design the primers?

2. Please specify the manufacturer of the Taq PCR master mix.

3. What thermocycler was used for qPCR?

Results

-          Figure 2 1. Exon 1 is incorrect in the title, exon 2 should be correct.

2. There is no description below the figure, there is no indication of the size range of the ladder.

3. The sizes of the PCR products in the figure do not correspond to the sizes listed in Table 1.

4. Heterozygotes are not marked - according to the sequence data, a heterozygote AG (simultaneous green and black peak) was recorded for each SNP - however, this is not highlighted in the sequence.

-          Table 7. - For the trait CLR, in the case of the H1 haplotype, no significance is given.

Author Response

Dear reviewer:

Thank you for your review and suggestions on our manuscript. Those comments are all valuable and very helpful for revising and improving our paper. We have studied comments carefully and have made correction which we hope meet with approval. The comments are laid out below in italicized font and specific concerns have been numbered. Our response is given in red font and changes/additions to the manuscript are given in the red text. We hope these changes make the manuscript acceptable for publication.

 Q1- 1. What software was used to design the primers.

Response:

Thank you for carefully reading our manuscript and giving us valuable suggestion.

We used primer5 software to design primers that have been added in the article.

 Q2-   Please specify the manufacturer of the Taq PCR master mix.

Response:

Thank you for carefully reading our manuscript and giving us valuable suggestion.

We have supplemented the manufacturer of Taq PCR manufacturer

Q3-   What thermocycler was used for qPCR?

Response:

Thank you for carefully reading our manuscript and giving us valuable suggestion.

We used Thermofish Scientific,Inc thermocycler for qPCR that have been added in the article.

  Q1 Figure 2 – 1. Exon 1 is incorrect in the title, exon 2 should be correct.

Corrected.

  1. There is no description below the figure, there is no indication of the size range of the ladder.

Corrected.

  1. The sizes of the PCR products in the figure do not correspond to the sizes listed in Table 1.

Thank you for carefully reading our manuscript and giving us valuable suggestion.

Corrected.

We have redesigned the primers and added primer information and pictures.

  1. Heterozygotes are not marked - according to the sequence data, a heterozygote AG (simultaneous green and black peak) was recorded for each SNP - however, this is not highlighted in the sequence.

Thank you for carefully reading our manuscript and giving us valuable suggestion.

Corrected

We have added sequence mutation sites highlighted in the image

Table 7. - For the trait CLR, in the case of the H1 haplotype, no significance is given.

Corrected.

Round 2

Reviewer 3 Report

Comments and Suggestions for Authors

After reading the responses to the comments and changes made in the text, I accept the explanations and corrections made by the authors.

Author Response

I would like to thank the editor and reviewers for their valuable comments on my manuscript, we have completed the revisions and sincerely hope that these revisions will improve the paper manuscript and will be published in this journal.